# Roles of Integrin in Cardiovascular Diseases: From Basic Research to Clinical Implications

**DOI:** 10.3390/ijms25074096

**Published:** 2024-04-07

**Authors:** Shuo Zhang, Qingfang Zhang, Yutong Lu, Jianrui Chen, Jinkai Liu, Zhuohan Li, Zhenzhen Xie

**Affiliations:** 1College of Basic Medical, Nanchang University, Nanchang 330006, China; 15806393806@163.com (S.Z.); zqf3196934718@163.com (Q.Z.); a18263861445@126.com (Y.L.); sdchenjianrui@163.com (J.C.); 19970410623@163.com (J.L.); 15355186901@163.com (Z.L.); 2Queen Mary School, Medical Department, Nanchang University, Nanchang 330031, China

**Keywords:** integrins, cardiovascular diseases, vascular endothelial cells, vascular smooth muscle cells, platelets, cardiac fibroblasts, cardiomyocytes, integrin antagonists and antibodies, nanotherapy

## Abstract

Cardiovascular diseases (CVDs) pose a significant global health threat due to their complex pathogenesis and high incidence, imposing a substantial burden on global healthcare systems. Integrins, a group of heterodimers consisting of α and β subunits that are located on the cell membrane, have emerged as key players in mediating the occurrence and progression of CVDs by regulating the physiological activities of endothelial cells, vascular smooth muscle cells, platelets, fibroblasts, cardiomyocytes, and various immune cells. The crucial role of integrins in the progression of CVDs has valuable implications for targeted therapies. In this context, the development and application of various integrin antibodies and antagonists have been explored for antiplatelet therapy and anti-inflammatory-mediated tissue damage. Additionally, the rise of nanomedicine has enhanced the specificity and bioavailability of precision therapy targeting integrins. Nevertheless, the complexity of the pathogenesis of CVDs presents tremendous challenges for monoclonal targeted treatment. This paper reviews the mechanisms of integrins in the development of atherosclerosis, cardiac fibrosis, hypertension, and arrhythmias, which may pave the way for future innovations in the diagnosis and treatment of CVDs.

## 1. Introduction

Cardiovascular diseases (CVDs) encompass a spectrum of disorders affecting the heart and vasculature, constituting a substantial global health burden [1]. In 2022, cardiovascular diseases were responsible for approximately 190,000 global fatalities [1]. An estimated 34% of these cardiovascular disease-related deaths occur before the age of 70 [1,2]. CVDs, which are classified according to affected sites and involve various cell types, encompass a spectrum of conditions including cardiomyopathy, arrhythmia, atherosclerosis (AS), cardiac fibrosis, heart failure, coronary artery disease, and hypertension [2,3]. Current research indicates that the interaction between cells and the extracellular matrix (ECM) plays a significantly pivotal role in the pathogenesis of CVDs [4,5,6,7]. Notably, specific focus has been directed toward dysregulated ligands that target integrins, which emphasizes their unique relevance in the context of cardioprotection [8]. Integrins, which are the principal cellular adhesion receptors for ECM constituents, comprise a family of 24 transmembrane heterodimers formed by a combination of 18 α and 8 β integrin subunits [9]. Integrins function as receptors for ECM components categorized into four distinct groups, including those containing the RGD (Arg-Gly-Asp) motif, leukocyte-specific ligands, collagen ligands, and laminin ligands (Figure 1) [10,11]. In each heterodimeric complex, there is a distinct extracellular domain, a transmembrane domain, and a concise cytoplasmic domain connected to the cytoskeleton (Figure 1) [9]. The ectodomain of the α subunit comprises four extracellular regions: a seven-bladed β-propeller, a thigh region, and two calf domains (Figure 1) [9]. The shared feature among various α subunits within their extracellular domains is the presence of seven repetitive motifs, which assemble into a seven-bladed propeller configuration on the top surface (Figure 1) [9]. Additionally, on the underside of blades 4–7, sites for binding divalent cations can be found (Figure 1) [9]. The β subunit’s ectodomain features seven domains with complex insertions: a β I domain within the hybrid domain, a plexin–semaphorin–integrin domain, four cysteine-rich epidermal growth factor modules, and a β tail domain (Figure 1) [9]. These structural components facilitate the initiation of both inside-out and outside-in signaling pathways, thereby intricately orchestrating cellular responses [12]. In the context of inside-out signaling, activated chemokines can propagate signals through G-protein-coupled receptors [11]. This signaling cascade initiates phosphorylation events within the cytoplasmic domain of the receptor subunit, thereby enhancing the affinity of integrins to their ligands [11]. This mechanism induces conformational changes in the extracellular domain of integrins, profoundly affecting critical biological processes, including cell adhesion, migration, proliferation, survival, and differentiation [11]. In the outside-in signaling cascade, ligand binding induces conformational changes in integrins situated on the plasma membrane, which leads to the separation of their heads from their tails [11]. This conformational alteration facilitates the interaction of the cytoplasmic tail domain with intracellular signaling molecules [11]. Consequently, a spectrum of cellular responses ensues, encompassing modifications in cell morphology and structure, the modulation of cell adhesion and migration, the regulation of cell proliferation and survival, and control over cellular differentiation [11].

During the progression of CVDs, integrins that are present on diverse cell types establish aberrant connections between the ECM and intracellular signaling pathways. This process triggers multiple pathophysiological mechanisms characterized by dysregulation, resulting in adverse responses, including inflammation, cellular dysfunction, and aberrant proliferation, migration, adhesion, and differentiation [13]. Significantly, the interaction between ECM components and integrins that are present on diverse cell types, including cardiomyocytes, vascular endothelial cells (VECs), vascular smooth muscle cells (VSMCs), and immune cells, can potentially either facilitate or impede the pathogenesis and progression of CVDs [14,15,16,17]. As a result, the modulation of integrin-mediated signaling pathways presents a promising way of attenuating the deleterious cascades associated with these diseases, thereby mitigating the risk of CVD development and progression [18,19].

This review delineates the pathophysiological dysregulation mechanisms triggered by interactions between the ECM and integrins, elucidating their intricate roles in the progression of prevalent clinical manifestations of CVDs. Specifically, a detailed examination is provided for AS, hypertension, arrhythmias, and cardiac fibrosis [16,20,21,22]. As the intricate pathogenic mechanisms orchestrated by integrins in CVDs have become more understood, integrin-targeted therapeutics have been clinically employed for CVD treatment. These therapeutics exhibit noteworthy integrin-targeting specificity, low toxicity profiles, and robust efficacy [23]. At present, there are two integrin-targeted therapeutic modalities in clinical practice for addressing CVDs: integrin antagonists and antibodies and integrin-directed nanotherapy [23,24]. As our understanding of the contributions of integrins to the pathogenesis of CVDs advances, it is increasingly evident that therapeutic strategies targeting integrins hold considerable promise for enhancing the management of CVDs.

## 2. Potential Effects of Integrins in CVDs

The involvement of different cell surface integrins plays a crucial role in the development and progression of CVDs. Figure 2 demonstrates the signaling pathways of integrins and related molecules in the six specific cell types that are mainly involved in the pathogenesis of CVDs.

### 2.1. Atherosclerosis

AS is a prominent contributor to global mortality and is characterized as a chronic inflammatory disease with the development of fibrofatty lesions on the arterial walls of large and medium-sized arteries [25,26]. This condition predisposes individuals to a spectrum of panvascular diseases, such as ischemic heart disease, stroke, and peripheral vascular disease, leading to a subsequent reduction in life expectancy [25,26]. AS exhibits a sophisticated formation mechanism that initiates with the deposition of low-density lipoprotein (LDL) in the endothelium of blood vessels [27,28,29]. LDL is subsequently translocated to the subendothelial space, where it undergoes oxidation, resulting in the formation of oxidized LDL (ox-LDL) [28,29]. Monocytes, macrophages, and smooth muscle cells actively phagocytize ox-LDL and further assume a foam cell phenotype, while fibroblasts synthesize and release fibrous tissue into the ECM, culminating in the establishment of a fibrous cap on the plaque surface [30,31,32,33]. Upon plaque rupture, platelets aggregate at the breach site and subsequently orchestrate the formation of a fibrin meshwork, which progressively develops to a thrombus [32,34]. Extensive research has underscored the presence of integrins on VECs, VSMCs, immune cells, and platelets, which play a pivotal role in the regulation of cellular activities in AS (Table 1).

#### 2.1.1. Integrins That Bind to RGD Receptors

The α5, α5β1, αVβ3, αVβ5, αVβ1, α8β1, and αIIbβ3 integrins located on the cell membranes of VSMCs, VECs, immune cells, and platelets are involved in the progression of AS. The excessive activation of α5 integrins on the cell membrane of ECs triggers inflammatory activation of the ECs and contributes to the development of atherosclerosis, and this activation can be hindered by the integrins directly binding to the C-terminus of cartilage oligomeric matrix protein [14]. Demos et al. reported that the activation of the calcium ion channel Piezo1 by disturbed blood flow and subsequent conformational changes in annexin A2 induces the translocation of α5 integrins on the plasma membrane of ECs to lipid rafts (Figure 2) [35,36]. The overwhelming impact of α5 integrins further leads to the activation of the endothelial inflammatory pathway, which mediates the engulfment of ox-LDL by monocytes (foam cells) and their deposition in the vascular intima, ultimately contributing to the development of atherosclerotic plaques (Figure 2) [35,36]. In addition, it has been elucidated that the overexpression of α5β1 integrins present in the ECM can also induce VSMC calcification, which drives the trans-differentiation of VSMC into osteochondrocyte-like cells and ultimately causes the deposition of calcium salts in the arterial wall and cardiac valves [58]. Zhu et al. indicated that the overexpression of α5β1 integrins present on VSMCs and the abnormal interaction between α5β1 integrins and α-smooth muscle actin (α-SMA) on VSMCs led to vascular tone dysregulation, resulting in altered contraction function, excessive migration, and the proliferation of VSMCs, ultimately resulting in AS (Figure 2) [40]. Previous research confirmed that the upregulation of α5β1 integrins on the VSMC cell membrane mediates the transition of VSMC from a “contractile” to a “synthetic” phenotype, which significantly contributes to VSMC proliferation and migration, thus playing a crucial role in the progression of AS [41]. It has been reported that the excessive activation of α5β1 integrins that are present on ECs by ox-LDL mediates the expression of endothelial pro-inflammatory genes, such as tumor necrosis factor (TNF)-α, IL-1β, and IL-6, which results in atherosclerotic inflammation (Figure 2) [59]. One study elucidated the critical role of α5β1 integrins on macrophages in promoting cell adhesion through excessive interaction with the immune cell-specific Eph receptor A2 [50]. Budatha et al. showed that α5β1 and αVβ3 integrins located on the plasma membranes of ECs are oversensitive to inflammatory stimuli when stimulated by fibronectin [60]. This sensitivity promotes microscopic changes during atherosclerosis, such as fibronectin deposition, the activation of EC inflammation, lipid accumulation, and thickening of the arterial wall [60]. Furthermore, it has been shown that the α5β1 and αVβ3 integrins located on the plasma membrane of VSMCs can be excessively activated by changes in the ECM, inducing the cytoplasmic translocation of focal adhesion kinase (FAK) and increasing the activity of proline-rich tyrosine kinase 2 (Pyk2), which promotes the rapid proliferation of VSMCs and plaque formation (Figure 2) [61]. Moreover, studies have demonstrated that the aberrant interaction between αVβ3 integrins on platelets and fibronectin contributes significantly to thrombosis within the injured vessel wall. This interaction is pivotal for platelet adhesion and aggregation on atherosclerotic plaques, thereby highlighting its importance in the pathophysiology of vascular events associated with atherosclerosis [56,57,62]. Okamoto et al. demonstrated that vascular stiffness and shear stress upregulate α5β1 integrins on the EC membrane and connexin in the ECM; the interaction of α5β1 integrins and connexin enhances vascular inflammation and leads to atherogenesis by promoting the activation of NF-κB and the expression of pro-inflammatory genes (Figure 2) [63]. Conversely, αVβ3 integrins and β1 integrins with talin-induced binding enhance the integrity of cell–cell junctions and protect endothelial barrier function [64]. Additionally, noteworthy studies highlighted that under pathological conditions, the levels of αV integrins on the surface of endothelial microparticles, which are vesicles derived from the EC membrane, increase and exert proinflammatory effects through their interaction with monocytes to promote atherosclerosis development [44]. Notably, the existing literature has consistently documented the involvement of αVβ5 integrins that are present on macrophages, which become activated through peroxisome proliferator-activated receptor gamma (PPARγ) [45]. This activation culminates in the polarization of macrophages towards the M2 phenotype with the subsequent expression of anti-inflammatory factors, such as interleukin-10 (IL-10) [45]. This orchestrated response promotes tissue repair and fosters clot formation, illustrating the multifaceted impact of integrins in ameliorating atherosclerotic processes [45]. Recent studies have reported the role of omentin-1, a newly discovered cytokine predominantly secreted by epicardial adipose tissue and characterized by a fibrinogen-like domain, that is upregulated in AS [46]. Omentin-1 has been shown to interact with αVβ5 integrins located on the cell membrane of macrophages, eliciting the downstream activation of the Akt/AMPK signaling cascade (Figure 2) [46]. This signaling axis exerts inhibitory effects on the formation of necrotic cores and the expression of pro-inflammatory cytokines within atherosclerotic lesions, thereby fostering plaque stability [46]. Extensive research has illuminated the dual effects of osteopontin (OPN) on a spectrum of integrins, including αVβ1, αVβ3, αVβ5, and α8β1 integrins [47]. The acute elevation of OPN levels confers a protective effect by mitigating vascular calcification and promoting favorable ischemic neovascularization. Conversely, chronic elevation in OPN levels is clinically correlated with an augmented susceptibility to major adverse cardiovascular events [47]. Furthermore, it has been reported that αIIbβ3 integrins exert considerable influence on platelet functionality, with pivotal roles in activation, migration, and secretion [54,55,65,66]. Research has elucidated that the bivalent interaction between αIIbβ3 integrins presenting on platelets and fibrinogen facilitates platelet aggregation, thereby contributing to thrombus formation [65]. For example, studies have demonstrated that reelin, a glycoprotein released by platelets, exerts an overamplifying effect on the αIIbβ3 integrin-mediated outside-in signaling pathway by activating the Ras-related C3 botulinum toxin substrate and Ras homolog family member A (RhoA) [54,66]. This augmentation facilitates the cytoskeletal reorganization of platelets, thereby enhancing thrombus formation [54,66]. In contrast, the activation of αIIbβ3 integrins has been shown to be inhibited by GNE 495 and PF 06260933 (both potent and selective MAP4K4 inhibitors), the primary outcome of which is mitigating platelet aggregation and clot retraction [55]. Furthermore, Chen et al. highlighted that β1 integrins located on the plasma membrane of VECs are upregulated in response to galectin-3 stimulation, leading to the activation of the β1-RhoA-JNK signaling pathway, thereby exacerbating ox-LDL-mediated vascular endothelial injury, promoting inflammation, and facilitating foam cell deposition (Figure 2) [37]. Research has shown that β1 integrins expressed on the plasma membrane of VSMCs promote the overproduction of matrix metalloproteinases 2 (MMP) through the regulation of milk fat globule-epidermal growth factor 8 (Figure 2) [15]. Ultimately, this triggers the redundant activation of transforming growth factor-β1 (TGF-β1), which is incorporated into atherosclerotic plaques, and induces osteogenic differentiation and biomineralization in VSMCs, facilitating vascular wall calcification and sclerosis (Figure 2) [15].

#### 2.1.2. Integrins That Bind to Laminin Receptors

The β4, α6β1, and β3 integrins that are present on ECs and VSMCs play roles in the inflammation and plaque formation of AS. Research has shown that β4 integrins on the plasma membrane of ECs activates a positive feedback signaling pathway by activating SRC and downstream NF-κB, the activation of which contributes to excessive transcriptional activity of β4 integrins; the activation of the pathway then upregulates the expression of endothelial inflammatory factors, leading to the development of AS (Figure 2) [38]. Furthermore, a similar mechanism has been reported in which the α6β1 integrins present on ECs bind to the EC-expressed matrix protein CCN1 to activate NF-κB, which forms a positive feedback loop with the CCN1 and α6β1 integrins and amplifies oxidative stress and inflammation in the arterial wall (Figure 2) [39]. Additionally, a previous study has confirmed that the β3 integrins located on the cell membrane of VSMCs can be overstimulated by thrombin and then interact with CD47 to mediate VSMC migration and proliferation (Figure 2) [41]. This process possibly leads to increased vascular wall thickness and the formation of subendothelial plaques, which promotes vascular restenosis following percutaneous transluminal angioplasty [41].

#### 2.1.3. Integrins That Bind to Leukocyte-Specific Receptors

The β2, αMβ2, αDβ2, β1, and α4β1 integrins expressed on the cell membranes of ECs, VSMCs, and immune cells participate in AS development. Krautter et al. highlighted that the overproduction of β2 integrins with galectin-9, which is secreted by VECs, leads to the activation and differentiation of monocytes [42]. This molecular interaction consequently amplifies the advancement of atherosclerotic plaque formation [42]. Previous research has underscored the interaction between β2 integrins expressed on neutrophils and endothelial cell adhesion molecules, including E-/P-selectins and ICAM-1, particularly on human umbilical vascular endothelial cells [51]. The excessive synthesis of β2 integrins causes increased interaction that activates the Syk/Src signaling pathway, leading to a heightened calcium flux within adherent neutrophils (Figure 2) [51]. Consequently, this cascade of events enhances vascular wall stiffening and exacerbates immune responses associated with AS [51]. In addition, the upregulation of β2 integrin expression on neutrophils is facilitated by the intracellular translocation of peptidyl arginine deiminase 4 triggered by the chemokine (C-X-C motif) ligand 1 protein [53]. This phenomenon significantly contributes to the adhesive characteristics of neutrophils [53]. A previous study suggested that β2 integrins on the plasma membrane of eosinophils, activated by P-selectin glycoprotein ligand 1 and P-selectin, upregulate and bind to ICAM-1, leading to the firm adhesion of eosinophils to the vascular endothelium [52]. In addition, an investigation has demonstrated that developmental endothelial locus-1, which is a secretory protein, modulates the interaction with αMβ2 integrins presenting on leukocytes, thereby impeding αMβ2 integrins from binding to their endothelial counterpart, ICAM-1 [67]. Consequently, this impediment attenuates the adhesion and subsequent recruitment of leukocytes to inflamed sites [67]. Furthermore, Cui et al. demonstrated that the oxidative derivative resulting from the interaction of docosahexaenoic acid and 2-(w-carboxyethyl)-pyrrole acts as a ligand, facilitating binding to αDβ2 integrins situated on M1 macrophages [43]. This interaction enhances the accumulation of macrophages within the ECM of peripheral tissues, thereby contributing to the development of AS [43]. Moreover, previous investigations have shown that fatty-acid-binding protein 4 (FABP4), recognized as a member of the adipocyte fatty-acid-binding protein family, functions as an intracellular fatty acid transporter within macrophages [48,49]. Notably, its expression is upregulated by ox-LDL in monocytes and human coronary endothelial cells (HCAECs) [48,49]. Extensive research has delineated that the binding of FABP4 to α4 integrins, which are expressed on the plasma membrane of monocytes and HCAECs, facilitates the adhesion of monocytes and HCAECs [48,49]. Furthermore, cytotoxic T lymphocytes were reported to exhibit elevated expression levels of α4β1 integrins in response to various cytokines [68]. This upregulation is associated with the release of inflammatory mediators, thereby facilitating immune cell recruitment and contributing to the formation of atherosclerotic plaques [68]. A study describing the IncRNA associated with the progression and intervention of AS (RAPIA) showed that the β1 integrin/microRNA-183-5p signaling pathway in macrophages is upregulated in the presence of RAPIA expression, consequently promoting the progression of AS [69].

### 2.2. Cardiac Fibrosis

Cardiac fibrosis is a common terminal stage in various CVDs, such as myocardial infarction (MI), hypertension, and heart failure [22,70,71]. It refers to the increased deposition of collagen and other molecules within the ECM, resulting in changes in cardiac structure and function [22,70,71]. Research has indicated that immune cells, including monocytes, macrophages, and neutrophils, mediate inflammatory progression and ECM remodeling, while fibroblasts synthesize collagen to promote collagen deposition in the ECM [72,73,74,75]. The pathogenesis of cardiac fibrosis mainly involves the actions of fibroblasts, which cause cardiac remodeling mediated by collagen synthesis and deposition, and therefore, this section has categorized and summarized the role of cardiac cells, primarily fibroblasts, based on different integrin types. Many studies have elucidated the presence of integrins on cardiac cells, cardiac fibroblasts, and immune cells, highlighting their critical involvement in the regulation of cellular activities associated with cardiac fibrosis (Table 2).

#### 2.2.1. αV Integrins

Numerous studies have shown that αV integrins play a pivotal role in the progression of cardiac fibrosis. They primarily participate in fibroblast proliferation, adhesion, migration, and differentiation and monocyte and macrophage migration and adhesion, thereby mediating cardiac inflammation and collagen deposition [72,73,78,83]. Moreover, αVβ5 and αVβ3 integrins on the cell membrane of human cardiac fibroblasts (CFBs), induced by mechanical stress from fibroblast contraction, demonstrate conformational changes and mediate the excessive activation and release of TGF-β1, promoting fibroblast differentiation into collagen-producing myofibroblasts in the ECM [84]. Furthermore, Olivier et al. highlighted that αV integrins facilitate the over-release of TGF-β from latent TGF-β stores by redundantly interacting with the RGD motif on latent TGF-β in the ECM [85]. Similarly, previous research indicated that the overexpression of αV integrins on the plasma membrane of CFBs bind to the RGD site of latency-associated peptide (LAP) and lead to the excessive activation and release of TGF-β1, promoting the proliferation and collagen secretion of CFBs and inflammatory responses, thereby further enhancing the development of cardiac fibrosis [76]. Furthermore, several studies have confirmed that the upregulation of αVβ3 and αVβ5 integrins on fibroblasts can significantly induce the release of TGF-β1 into the ECM and further activate the TGFβ1/Smad3 pathway through autocrine signaling to enhance the expression of α-SMA, which facilitates fibroblast trans-differentiation and the synthesis of type I collagen, ultimately resulting in cardiac fibrosis-induced heart dysfunction (Figure 2) [77,86]. In addition, Patrick et al. indicated that excessively stimulated αVβ5 integrins present on CFBs bind to FAK, an integrin-associated protein, to promote the overproduction of α-SMA and type I collagen, the result of which is associated with myofibroblast transformation and collagen deposition (Figure 2) [87]. Investigations have demonstrated that overactivated αVβ3 integrins on CFBs bind to ECM proteins containing the RGD motif to activate c-Src, Pyk2, and FAK downstream, which mediate the adhesion, spreading, proliferation, and migration of fibroblasts and induce the accumulation of interstitial fibronectin and collagen within the ECM (Figure 2) [78]. Similarly, Chen et al. demonstrated that αVβ3 and α8β1 integrins, located on the cytoplasmic membrane of VSMCs and myofibroblasts, can be upregulated by angiotensin II, ultimately facilitating collagen contraction [73]. Additionally, the findings revealed that the upregulated αVβ3 integrins located on the plasma membrane of CFBs facilitate the αVβ3 integrin-dependent excessive activation of pro-MMP2 in aortic smooth muscle cells (ASMCs), and the activated MMP2 can degrade the ECM, which results in the breakdown and remodeling of constitutive proteins, such as fibronectin and collagen [88]. In addition, αV integrins regulate the function of immune cells and cytokines. The αVβ3/FAK-Src/NF-κB signaling pathway is utilized by TN-C (an ECM protein) to accelerate macrophage migration and the synthesis of proinflammatory/profibrotic cytokines, thereby enhancing the inflammatory response and increasing the synthesis of collagen I by fibroblasts, which consequently advances fibrotic progression [73,89]. Previous research highlighted that the αVβ3-dependent activation of microfibrillar-associated protein 4 (MFAP4) facilitates monocyte migration and directly promotes the MMP9-mediated degradation of the vascular wall induced by TNF [83]. Furthermore, αVβ3 integrins located on the plasma membrane of ASMCs have been shown to stimulate the increased proliferation of CD25+ FoxP3+ Treg cells and IL-10 expression, thereby mediating the recruitment of inflammatory cells and the production of pro-inflammatory cytokines [88]. Moreover, the upregulation of platelet-derived growth factor receptor beta (PDGFRβ), which is dependent on αV integrins in mesenchymal cells, can induce cardiac fibrosis [90]. It has been shown that the αV/β5-AKT signaling pathway of cardiomyocytes in type 2 diabetes stimulates the apoptosis of cardiomyocytes and myocardial fibrosis [91]. Additionally, research has elucidated that MFAP4, located in the connective tissue of the aortic wall, binds to RGD and excessively interacts with αVβ3 integrins located on the plasma membrane of VSMCs to transmit pro-fibrotic ECM signals to VSMCs, thus promoting the phenotypic conversion, dedifferentiation, proliferation, and migration of VSMCs [83].

#### 2.2.2. β1 Integrins

β1 integrins can produce a positive effect on the activation, proliferation, and differentiation of fibroblasts, as well as the synthesis and contraction of collagen. The overexpression of αVβ1 integrins on the cell membrane of fibroblasts has been shown to stimulate TGF-β1 expression and αVβ1-mediated FAK-Akt-mTOR and TGF-β receptor-mediated Smad2/3 signaling pathways to promote the internal transcription of α-SMA in fibroblasts, which induces the transition of fibroblasts to myofibroblasts (Figure 2) [72,92,93,94,95]. Similarly, previous research highlighted that the overactivation of β1 integrins on the fibroblast plasma membrane by ECM proteins, such as fibronectin, triggers the expression of phosphorylated FAK protein and α-SMA, ultimately promoting the migration of fibroblasts (Figure 2) [80]. β1 integrins mediate myocardial fibrosis via TGF-β signaling, with the participation of Smad2/3, a-smooth muscle actin, and collagen I (Figure 2) [79]. Previous research showed that α5β1 integrins present on the cell membrane of myofibroblasts induce the overproduction of fibronectin to mediate the further differentiation of myofibroblasts, which stimulates myofibroblasts to synthesize and secrete more collagen to enhance the deposition and reconstruction of the ECM [96]. It has been shown that α1β1 and α3β1 integrins can also promote myofibroblast differentiation and redundant collagen synthesis [96]. Leask et al. indicated that the upregulation of α11 and β1 integrins present on the cell membrane of myofibroblasts can very strongly activate downstream signaling pathways, including FAK, TGF-β-activated kinase, and reactive oxygen species by identifying high-stiffness ECM, resulting in the increased production, adhesion, and contraction of the ECM [74]. Furthermore, a recent study demonstrated that the overexpression of β1 integrins on the cytoplasmic membrane of fibroblasts activate PDGFRβ redundantly by acting on small proline-rich repeat 3 (SPRR3), which then crosstalks with PDGFRβ to regulate the Akt, FAK, ERK, and p38 signaling pathways to enhance fibroblast proliferation, collagen expression, and interstitial fibrosis [97]. Additionally, it has been reported that the overexpression of β1 integrins present on the plasma membrane of fibroblasts can identify stiffness in the ECM after MI by recognizing collagen accumulation in fibrotic tissues and triggering intracellular signaling in fibroblasts, which express vestigial-like family member 3, enhancing collagen synthesis [98]. Moreover, a previous study highlighted that the increasing association of β1 integrins and CD63 (a tissue inhibitor of metalloproteinase-1 (TIMP1) cell surface receptor) on CFBs is mediated by TIMP1 to initiate the activation and nuclear translocation of Smad2/3 and β-catenin, contributing to de novo collagen overproduction [99]. Upregulated αV integrins on the cytoplasmic membrane of cardiac PW1+ stromal cells, a type of cell that can differentiate into many cell types of the heart and is related to regeneration and repair, activate TGF-β on the surface of the cells, thus promoting the overexpression of fiber genes in cardiac PW1+ stromal cells and mediating PW1+ cells to differentiate into fibrocytes after MI [100]. In addition, it has been reported that the excessive interactions between α2β1 integrins on CFBs and type I collagen regulate the overproduction of lysyl oxidase, which catalyzes the crosslinking between collagen and elastin to form a more stable and compact network of collagen fibers, thereby increasing the persistence of fibrotic regions [101]. Furthermore, recent research has found that α7β1 integrins located on the cardiomyocyte cytoplasmic membrane are laminin receptors that facilitate laminin deposition in the cardiomyocyte lamina [102].

β1 integrins also have inhibitory effects on cardiac fibrosis. Combinations of different α subunits of β1 integrins lead to different types of integrins involved in different signaling pathways. Furthermore, α2β1 integrins located on the plasma membrane of fibroblasts recognize stiff ECM and trigger the downstream signaling pathways that induce FAK and Src kinase to inhibit collagen accumulation in human atrial fibroblasts [81]. Hong et al. indicated that α2β1 integrins on the cytoplasmic membrane of myofibroblasts regulate their downstream target PTEN through excessive crosstalk with PP2A and promote the overactivation of PETN, thus over inhibiting the activation of AKT and the expression of α-SMA, which suppresses the differentiation and proliferation of myofibroblasts (Figure 2) [82]. Additionally, α2β1 integrins present on the plasma membrane of fibroblasts can induce the upregulation of MMP1 to prevent collagen accumulation in the ECM, thereby protecting cardiomyocytes from fibrosis (Figure 2) [96]. Notably, we found that α2β1 integrins both promote and inhibit cardiac fibrosis. This was related to the different sources of cells (e.g., mice or aortic stenosis patients) and different modeling methods (e.g., mouse aortic ligation, gene editing, or fibrosis induction factor cultured cells).

#### 2.2.3. Other Integrins

Several studies have concluded that other types of integrins can also play an important role in the development of cardiac fibrosis. Moreover, α11 integrins present on the cell membrane of fibroblasts can stimulate Smad2/3 in the promoter of α11 integrins by a TGF-β-dependent mechanism to promote the overexpression of α11 integrins, thereby mediating myofibroblast formation and extremely high type I collagen deposition [103]. Similarly, previous studies highlighted that α11, α5, β3, and α2 integrin synthesis is accelerated by the binding of TGF-β to its receptors on fibroblasts, which activates fibroblast trans-differentiation into myofibroblasts to synthesize type I and III collagens [104,105]. It has been reported that the expression of α11 integrins in patients with diabetes can be induced by the TGF-β2 autocrine signaling pathway in fibroblasts, which is triggered by glycated collagen in the cardiac stroma [106]. Previous research showed that α5 integrins on the plasma membrane of fibroblasts bind to miR-30d expressed by cardiomyocytes to inhibit fibroblast proliferation and activation, thereby inhibiting pathological left ventricle remodeling and fibrosis [107]. In addition, Liu et al. indicated that the high transcription of β2 integrins in macrophages is coordinated by myocardin-related transcription factor A and the transcription factor Sp1, contributing to the adhesion and migration of macrophages, thus promoting inflammation and ECM remodeling [75]. Upregulated αMβ2 integrins on the neutrophil plasma membrane can facilitate the increasing infiltration of neutrophils into atrial tissue, thereby facilitating the development of an inflammatory response and atrial fibrosis [73]. Furthermore, it is believed that αIIbβ3 integrins on platelets increase their expression upon the stimulation of 5-HT released from activated platelets, enhancing binding between platelets and fibronectin and the coagulation process, and they participate in inflammatory response and tissue repair, which ultimately contributes to the development of cardiac fibrosis [108]. A previous study elucidated that the upregulation of α4 integrins located on the cell membrane of type 1 T helper cells (Th1) mediate the increasing adhesion of Th1 cells in an IFN-γ-dependent manner and induce the over release of TGF-β from CFBs and the transition of CFBs to myofibroblasts [109]. It has been reported that the excessive activation of α4 integrins present on the cell membranes of monocytes and macrophages mediates the increased adhesion and infiltration of monocytes and macrophages to cardiac tissues, and monocytes can transform into macrophages that release inflammatory mediators and MMPs under inflammatory stimuli, thereby participating in the progression of cardiac inflammation and ECM remodeling [110].

### 2.3. Arrhythmias

Arrhythmias represent a significant health burden and are characterized by disruptions in cardiac rhythm [17,20,111,112,113,114,115,116,117]. They often stem from diverse cardiomyopathies, including valvular heart disease, arrhythmogenic cardiomyopathy (ACM), arrhythmogenic right ventricular cardiomyopathy, and cardiac aging [17,20,111,112,113,114,115,116,117]. The pathogenesis of arrhythmias involves myocardial damage, dysfunction in myocardial electrical signaling, and cytoskeletal remodeling [17,20,111,112,113,114,115,116,117]. Extensive research has underscored the pivotal role of integrins, which are prominently expressed on cardiomyocytes, in the pathogenesis of arrhythmias (Table 3). 

Integrins play a crucial role in cytoskeletal remodeling and other essential biological processes through their interaction with the ECM [20]. The β1 integrin/integrin-linked kinase (ILK) signaling pathway has been reported to exert a dual regulatory function in the modulation of cardiac aging [20]. This pathway is implicated in the colonization proximal to cardiomyocyte contacts and Z-bands, which are pivotal for the establishment and preservation of the structural integrity of the heart [20]. A slight downregulation of the β1 integrin/ILK signaling pathway has been demonstrated to attenuate age-related arrhythmia deterioration and prolong lifespan [20]. In other investigations, β1 integrins have been proposed to phosphorylate connexin-43 (Cx43), which is the predominant constituent of the gap junction channels found on cardiomyocytes (Figure 2) [111,112,113]. This phosphorylation event is believed to occur via the aberrant activation of the ILK/Akt/Cx43 signaling pathway (Figure 2) [111,112,113]. Consequently, it is suggested that this molecular mechanism attenuates cardiac remodeling and mitigates the occurrence of ischemia/reperfusion-induced ventricular arrhythmias [111,112,113]. Additionally, vinculin is localized at cell-to-cell intercalated discs and cell-to-matrix adhesion sites (Figure 2) [17]. It functions as an actin-binding protein, organizing and anchoring the actin cytoskeleton to the cardiomyocyte membrane via β1 integrin-containing focal adhesions (Figure 2) [17]. This anchoring mechanism contributes to the maintenance of a stable gap junction containing Cx43 [17]. However, vinculin is suppressed in ACM, resulting in cardiac fibrosis and arrhythmias [17]. Furthermore, contemporary investigations have highlighted that severe ventricular arrhythmias provoked by RyR2 malfunction represent notable features of arrhythmogenic right ventricular cardiomyopathy, an inherited desmosomal cardiomyopathy [114]. The absence of mutations in desmosomal genes (DSC, DSG, and DSP) leads to elevated fibronectin synthesis and secretion, promoting fibronectin binding to β1 integrins and their subsequent degradation (Figure 2) [114]. The reduction in β1 integrins present on the cell membrane of cardiomyocytes results in an upregulation of RyR2 Ser-2030 phosphorylation levels, leading to a destabilization of the RyR2 protein structure (Figure 2) [114]. Consequently, this disruption contributes to intracellular calcium dysregulation, manifesting as calcium leakage, which, in turn, triggers ventricular tachycardia (Figure 2) [114]. Furthermore, calreticulin, an endoplasmic reticulum calcium-buffering protein, has been found to play a role in mediating atrial fibrillation in valvular heart disease (Figure 2) [115]. This involvement occurs through the excessive activation of the TGF-β1/fibronectin/α5 integrin signaling pathway in fibroblasts, leading to heightened fibroblast proliferation and dysregulated ECM deposition within the cardiac interstitium (Figure 2) [115]. In the pathogenesis of arrhythmias observed in ACM that result from genetic mutations affecting the desmosomal cell–cell adhesion complex, particularly the desmoglein-2-W2A mutation, it has been reported that aberrantly activated αVβ6 integrins engage with LAP, leading to the dissociation of the TGF-β1-LAP complex [116]. Subsequently, increasing the TGF-β1 levels initiates a profibrotic downstream signaling cascade mediated through SMAD molecules [116]. The subsequent outcomes encompass fibrofatty deposition and potentially fatal arrhythmia [116].

### 2.4. Hypertension

As a widespread contributor to cardiovascular mortality and morbidity, hypertension entails pathological ECM–integrin interactions [21,118]. This contributes to a spectrum of biological effects, including increased blood vessel rigidity, the migration and proliferation of ASMCs, EC dysfunction, and the formation of neointima within the vascular wall [21,118]. Multiple studies have revealed the presence of integrins on ASMCs and ECs, highlighting their crucial role in regulating cellular activities linked to hypertension (Table 4).

Drawing on pulmonary arterial hypertension (PAH) as an illustrative example, extensive documentation exists regarding the interaction between β3 integrins on the cell membrane of pulmonary artery smooth muscle cells (PASMCs) and MMP8, which is secreted by PASMCs (Figure 2) [119]. This interaction subsequently orchestrates the activation of the FAK/Yes-associated protein/transcriptional coactivator with the PDZ-binding motif signaling pathway, thereby facilitating PASMC proliferation and instigating vascular remodeling in the context of PAH (Figure 2) [119]. Furthermore, hypoxia has been identified as a critical factor in the pathogenesis of PAH [120]. It is implicated in the aberrant activation of the α5β3 integrin/Pyk2/ERK/NF-kB/Nox4/H_2_O_2_ signaling cascade in PASMCs, which subsequently downregulates peroxisome PPARγ expression (Figure 2) [120]. This downregulated PPARγ expression is associated with the promotion of PASMC proliferation [120]. Similarly, hypoxia can also mediate the activation of the α5β3 integrin/osteoprotegerin/FAK signaling pathway, leading to PAH (Figure 2) [61,121]. Osteoprotegerin, belonging to the tumor necrosis factor receptor superfamily and expressed in cardiac tissue under hypoxic conditions, has been investigated for its potential contribution to PAH [121]. This is postulated to occur through the aberrant activation of the αVβ3 integrin/FAK/AKT signaling pathway in PASMCs, ultimately leading to the proliferation of PASMCs (Figure 2) [121]. Furthermore, numerous studies have found that αVβ3 integrins located on the cytoplasmic membrane of PASMCs can bind to OPN, a pivotal contributor to the activated phenotype of PASMCs in hypoxic PAH [122]. Subsequently, this interaction mediates the aberrant activation of the ERK1/2 and AKT signaling pathways, culminating in the proliferation of PASMCs (Figure 2) [122]. In addition, these findings revealed that the expression of β5 integrins on the cell membrane of PASMCs is abnormally upregulated by platelet-derived growth factor beta polypeptide b (PDGF-BB) stimulation [123,124]. Elevated levels of β5 integrin aberrantly activate dual pathways [123,124]. β5 integrins engage with miR-96-5p, consequently suppressing its activity [123]. The diminished expression of miR-96-5p results in the upregulation of mTOR levels, thereby fostering a synthetic phenotype of PASMCs [123]. Alternatively, β5 integrins upregulate the expression of the Uba1 protein, activating Ube2n and Mdm2 (Figure 2) [123,124]. This cascade of events leads to the downregulation of angiotensin-converting enzyme 2 expression and the induction of a synthetic phenotype in PASMCs (Figure 2) [123,124]. Consequently, this molecular mechanism contributes to pulmonary artery remodeling and the pathogenesis of PAH [123,124]. In the context of PAH, the progression of vascular stiffness and the consequential loss of elastic properties are attributed to the involvement of the overactivation of β5 integrins [125]. β5 integrins have been demonstrated to promote barrier functions while inhibiting the endothelial-to-mesenchymal transition gene expression induced by TGF-β [125]. In addition, a plethora of studies have underscored the interaction between upregulated α5β1 integrins located on VSMCs and fibronectin, leading to VSMC adhesion [40,60,126]. Furthermore, fibronectin has been implicated in fostering inflammation by engaging α5β1 integrins on VECs, culminating in vascular remodeling and hypertension as observed in subsequent outcomes [40,60,126]. Consistently, recent research showed that the aberrant activation of αVβ5 integrin triggers the TGF-β/SMAD2/3 signaling pathway (Figure 2) [86]. This aberrant activation, in turn, promotes the upregulation of α-SMA and facilitates the differentiation of fibroblasts into contractile myofibroblasts (Figure 2) [86]. Consequently, this process contributes to the induction of ECM stiffness, which has been associated with the pathogenesis of hypertension [86]. Apart from these findings, research has revealed that pregnancy-induced hypertension represents a notable form of hypertension attributable to a deficiency in activin A [127]. This deficiency exerts its influence by suppressing the ActRII/ALK4/SMAD2/3/4 signaling pathway, subsequently upregulating β1 integrin expression on trophoblasts [127]. Consequently, this suppresses trophoblast invasion into the uterine wall tissue, ultimately culminating in vascular remodeling and the manifestation of pregnancy-induced hypertension [127].

## 3. Integrin-Based Therapy

The development of integrin-based therapies for CVDs is progressing slowly, with the majority of therapeutic strategies still in the research phase. Presently, two main directions for integrin therapy exist. The first involves the use of integrin antagonists or antibodies to block the binding of integrins with ligands, thereby inhibiting signaling pathway transmission and disease progression [8,19,24]. The second involves exploiting the binding mechanism between integrins and ligands to fabricate nanoparticles (NPs) for drug encapsulation, aiming to enhance targeting specificity and bioavailability, delay drug release, and mitigate drug toxicity [128,129,130,131,132]. Figure 3 and Table 5 summarize the application of integrins in these two therapeutic approaches.

### 3.1. Integrin Antagonists, Antibodies, and Inhibitors

It has been demonstrated that αIIbβ3 integrins, situated on the surface of platelets, play a crucial role in mediating platelet binding to fibrinogen, thus promoting platelet spreading, aggregation, clot retraction, and thrombus stabilization (Figure 3) [18]. Consequently, αIIbβ3 integrin antagonists are employed in anti-thrombotic therapy for CVDs [18]. Numerous studies have demonstrated that αIIbβ3 integrin antibodies, including abciximab, and αIIbβ3 integrin inhibitors, including eptifibatide (RGD-mimetics) and tirofiban (RGD-mimetics), inhibit platelet binding to fibrinogen (Figure 3) [24,133,134,135,136,137,158]. These drugs treat acute coronary syndromes, including unstable angina and MI, and reduce the risk of bleeding after percutaneous coronary intervention compared to traditional antiplatelet therapy [24,133,134,135,136,137,158]. In addition, Hind et al. indicated that α2β1 integrin antagonists can suppress thrombosis by inhibiting the adhesion of platelets to collagen and the expression of collagen and collagenase genes (Figure 3) [134]. Research has shown that kallistatin, a unique serine proteinase inhibitor, can bind to β3 integrins to inhibit NF-κB nuclear translocation and downregulate vascular endothelial growth factor expression, thus suppressing angiogenesis and the progression of atherosclerotic plaques (Figure 3) [154]. Furthermore, recent research highlighted that antibodies against β2 integrins suppress leukocyte extravasation mediated by binding between leukocytes and ICAM-1 to reduce inflammatory tissue damage, thus treating AS development and plaque progression, MI, reperfusion injury, valvular stenosis, and cardiomyopathy (Figure 3) [8]. Additionally, it has been reported that antibodies against α2β1 integrins (vatelizumab) inhibit the transition of VSMCs from a “contractile” to a “synthetic” phenotype, which is known as SMC phenotypic modulation, and promote the aberrant proliferation, differentiation, and migration of VSMCs mediated by α2β1 integrins, thereby suppressing the formation of atherosclerotic plaques [142,159,160]. Similarly, the antibodies and antagonists of α4β1, α5β1, α9β1, αVβ3, and αVβ5 also restrain the chemotaxis of VSMCs and neointimal hyperplasia to limit the expansion of plaques and platelet aggregation (Figure 3) [141,142,144,145,146,147,150,151,152,153].

### 3.2. Nanotherapy

Nanomedicine has the potential to overcome certain obstacles faced by traditional medicine. Some of its advantages include high stability, high drug-carrying capacity, and various drug delivery modalities [155,161,162]. Overall, nanodrugs can be used in many ways to complement current therapeutic regimens, and the further exploration of NPs in CVDs will allow nanomedicine to reduce the global burden of CVDs. Several studies have reported the use of RGD-peptide, which specifically binds to the αIIbβ3 integrin ligand, as a surface molecule on nanocarriers for loading thrombolytic drugs. This enables targeted drug delivery to sites of platelet aggregation, thereby reducing the bleeding and toxicity risks associated with systemic administration (Figure 3) [19,155]. The selective inhibition of outside-in signals shows potential for inducing robust antithrombotic effects while mitigating the risk of bleeding complications [23]. The use of nanocarriers containing the Gα13-binding ExE motif within the integrin β3 cytoplasmic domain (M3mP6 or M3mp13) selectively targets the intracellular and extracellular signaling pathways of αIIbβ3 integrin, which effectively prevents secondary thrombus expansion mediated by outside-in signaling, subsequent vascular occlusion, and intravascular thrombosis, while allowing for primary platelet adhesion and aggregation, and was thus applied in the treatment of acute coronary syndrome and thrombotic cardiovascular events (Figure 3) [23,129,156]. Similarly, Linsey et al. demonstrated that ανβ3 integrins located on the surface of ECs are targeted by NP-wrapping cRGDfK peptides to deliver drug-containing NPs to atherosclerotic plaques (Figure 3) [130]. In addition, αVβ3 integrin-targeted and cathepsin k (CTSK)-responsive NPs are employed for the localized controlled release of the anti-inflammatory drug rapamycin (Figure 3) [157]. Rapamycin-targeted and responsive NPs (RAP@T/R NPs), containing the targeting polymer PLGA-PEG-c and the CTSK-sensitive polymer PLGA-Pep-PEG, accelerate rapamycin release in response to CTSK stimulation, leading to the significant inhibition of inflammatory macrophage uptake of ox-LDL and cytokine release, thereby suppressing systemic and local inflammation (Figure 3) [157]. Additionally, it is reported that α4β1 integrin-modified macrophage membrane NPs promote their accumulation in atherosclerotic plaques by binding to VCAM-1 on ECs [128].

## 4. Discussion

The current understanding of the mechanisms underlying CVDs revealed by research remains incomplete. Thus, a more in-depth exploration of the molecular mechanisms and associations between integrins and other CVDs that are not explicitly mentioned in this paper is required. Looking forward, CVDs increasingly stand as one of the most serious public health challenges, posing a significant threat to global health [163,164]. Abundant research indicates that integrins, as ubiquitous transmembrane heterodimeric proteins, are widely distributed on the cell membranes of VSMCs, ECs, fibroblasts, platelets, immune cells, and cardiomyocytes, and they play a crucial role in mediating cell signaling and physiological functions within the cardiovascular system [9]. Through integration with various cellular processes, integrins become indispensable in regulating cell functions such as inflammation, proliferation, migration, adhesion, and differentiation. The dysregulation of integrin signaling and conduction can lead to a variety of pathophysiological mechanisms, ultimately contributing to the initiation and progression of CVDs [165]. In the context of AS, integrins can orchestrate plaque formation by promoting EC oxidative stress and inflammation activation, recruiting and promoting the proliferation of VSMCs, inducing calcification, and allowing for the infiltration of immune cells and platelets [15,35,44,62]. Moreover, in the progression of cardiac fibrosis, integrins play a pivotal role in modulating inflammation driven by immune cells, the remodeling of the ECM, and the synthesis of collagen, thus exerting a significant impact on fibrosis development [75,85,108]. Additionally, in the realm of arrhythmias, integrins are implicated in the regulation of myocardial injury, dysfunction in cardiac electrical signaling, and the reconstruction of the cellular cytoskeleton, establishing a close association with the occurrence of rhythm disorders [20]. Furthermore, integrins contribute to the cellular and molecular mechanisms underlying the development of hypertension, participating in the regulation of increased vascular stiffness, VSMC migration and proliferation, EC dysfunction, and the formation of neointima within the vascular wall [21,118]. Notably, the long-term loading of Na+ in high-sodium-salt-diet-sensitive hypertension induces the destruction and reduction in glycocalyx (a protein covering the VEC membrane as a sensor of blood flow shear stress), which affects the excitation–secretion coupling of VECs, the excitation–contraction coupling of VSMCs, and the cellular homeostasis of Ca^2+^, Na^+^, and reactive oxygen species, ultimately leading to endothelial damage [166]. Salt-sensitive hypertension and the remodeling of human vascular smooth muscle cells induced by high sodium levels permanently reshape the sensitivity of the cells to short-term increases in circulating sodium levels following normalization at the systemic level [167]. Given the complexity of the mechanisms, unraveling the complex network of molecular interactions and pathways associated with CVDs holds tremendous potential to deepen our understanding of these complex diseases. Through an in-depth investigation into the relevant mechanisms, researchers can reveal the potential role of integrins in other types of CVDs, expanding our current knowledge base. Additionally, investigating the complex interplay between integrins and various types of CVDs beyond those explicitly mentioned in this review may lead to the discovery of new therapeutic targets and treatment strategies. Researchers can identify shared pathways and mechanisms by studying the intricate interactions between integrins and different types of CVDs, providing targets for developing innovative therapies and interventions. Therefore, due to the incomplete understanding of the potential mechanisms underlying CVDs, a more comprehensive exploration of molecular complexity and unexplored associations between integrins and other CVDs is urgently needed. Conducting such scientific investigations could potentially pave the way for breakthroughs in the diagnosis, prevention, and treatment of CVDs, ultimately improving the overall health outcomes of future patients.

A broad potential for integrin-related therapeutic approaches becomes evident given the impact of integrins on CVDs. Current research primarily concentrates on the treatment of AS, employing two main modalities—antibodies or antagonists and nanotherapy [8,129]. Integrin-targeting antibodies and antagonists, which include agents countering platelet and VSMC proliferation and migration, can inhibit integrin-mediated pro-plaque effects, thereby impeding AS progression [18,159]. Antiplatelet therapy continues to be the standard treatment for thrombotic diseases in the heart, peripheral arteries, and cerebral arteries with atherosclerotic thrombosis [135]. Based on the molecular mechanisms of thrombosis, three αIIbβ3 antagonists are currently used clinically, including eptifibatide and tirofiban from snake venom disintegrins and abciximab [135]. While various oral integrin αIIbβ3 antagonists have undergone testing, clinical trials have been halted due to adverse effects, such as high mortality rates or the lack of improvement in existing treatments [135]. Future research directions may involve structure–function analyses based on exogenous disintegrins, aiding in identifying platelet antagonists that maintain hemostasis while minimizing adverse effects [135]. Additionally, anti-VSMC antibodies and antagonists, such as vatelizumab and AJM300, can achieve therapeutic effects by inhibiting the integrin-mediated transition of VSMCs from a contractile phenotype to a synthetic phenotype, thereby suppressing VSMC proliferation, migration, and differentiation, ultimately blockading plaque formation [142,159,160]. Furthermore, α7β1 and α8β1 integrins prevent VSMC proliferation and migration, thus inspiring the development of integrin agonists, although currently, no relevant integrin-targeted drugs have been reported [168,169,170,171]. In addition, nanotherapy, a treatment method that uses NPs to deliver drugs, offers a unique approach. Unlike integrin antibodies and antagonists, nanotherapy not only possesses a targeting effect but also enhances targeting specificity and bioavailability, delays drug release, and mitigates drug toxicity [131,132]. Several studies have demonstrated that nanotherapy can be applied in the treatment of acute coronary syndromes and thrombotic cardiovascular events [23,129,156]. However, anti-atherosclerotic nanotherapies have undergone preclinical testing but have not yet been applied in clinical settings [172].

Current integrin-based therapeutic approaches are single-target integrin therapies. Future research should focus on developing multi-target therapies addressing diverse signaling pathways, which may yield superior therapeutic outcomes. Moreover, integrin-targeted drugs for other CVDs are promising and require further development. Additionally, the predominant treatment paradigm for contemporary CVDs continues to center on post-onset management and correction, exemplified by interventions such as thrombolysis, vasodilators, percutaneous coronary intervention, and coronary artery bypass grafting [173,174]. However, these approaches fail to address the underlying etiology and are therefore categorized as palliative treatments. Whether future efforts should concentrate on preventive treatments is a potential avenue for exploration, as indicated by our summary of integrin-related targeted therapies. Chinese scholars have proposed the concept of panvascular medicine, which provides a new perspective for the management of CVDs [174]. Panvascular diseases, characterized by vascular lesions and predominantly AS in 95% of cases, primarily affect vital organs, including the heart, brain, kidneys, limbs, and major arteries [174]. Panvascular medicine considers the systemic vasculature as a unified entity, explores the mechanisms underlying the development and progression of CVDs from a holistic perspective that encompasses both structural and functional aspects, and provides unified management of diseases [174]. The management of CVDs encompasses four aspects: prevention, diagnosis, treatment, and prognosis [174]. This includes targeting risk factors for vascular diseases, identifying more precise biomarkers, developing diagnostic tools or techniques with higher sensitivity and specificity (computed tomography angiography, magnetic resonance angiography, and ultrasound), developing more potent and compliant medications, strengthening patient follow-up, and organizing and summarizing data to further guide research and clinical practice [174]. Building upon this foundational concept, exploring the intricate relationship between integrins and CVDs could help catalyze advancements in early diagnostic processes and therapeutic interventions. Consequently, integrins could be strategically leveraged as pivotal biomarkers or therapeutic targets, thereby facilitating the early identification and management of cardiovascular conditions.

## Figures and Tables

**Figure 1 ijms-25-04096-f001:**
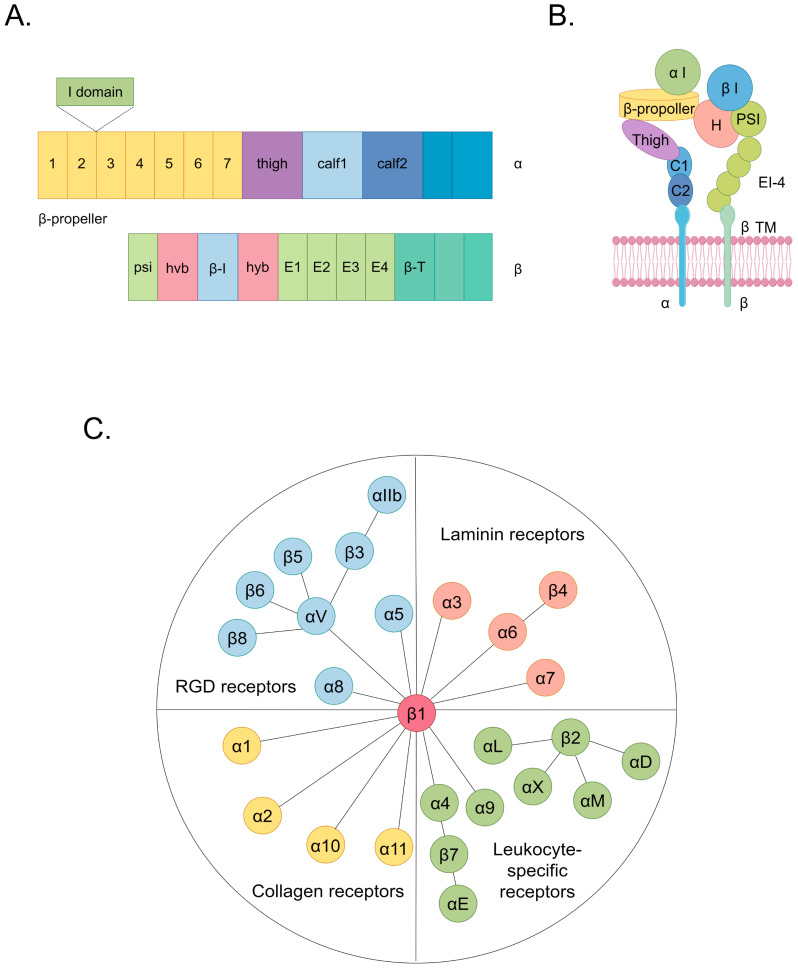
The structure and classification of integrins. (**A**) The domain organization of integrins within primary structures. (**B**) The spatial arrangement of domains within the representative three-dimensional crystal structure of integrins. (**C**) Integrins include 24 types. They can be classified into 4 categories based on their receptor specificity: RGD receptors, laminin receptors, leukocyte-specific receptors, and collagen receptors. The same color in (**A**,**B**) indicating integrin structure corresponds to the same domain. Among them, the hvb and hyb domains in (**A**) correspond to the H domain in (**B**).

**Figure 2 ijms-25-04096-f002:**
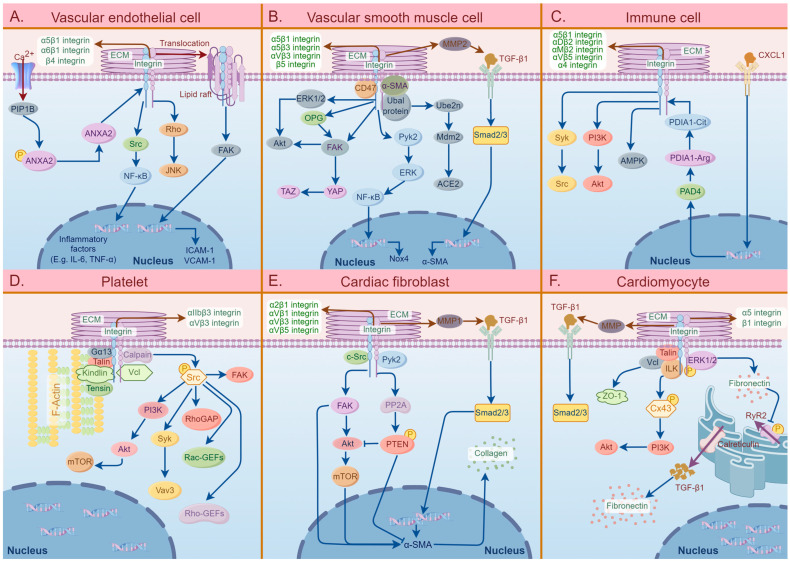
A summary diagram of the effects of integrins in six specific cell types that are mainly involved in the pathogenesis of cardiovascular diseases, including vascular endothelial cells (VECs), vascular smooth muscle cells (VSMCs), immune cells, platelets, cardiac fibroblasts, and cardiomyocytes. (**A**) The effects of integrins in VECs involved in the pathogenesis of atherosclerosis (AS) and hypertension. The extracellular matrix (ECM)–integrin interactions mediate VEC activation, dysfunction, and oxidation, thereby inducing vascular endothelial inflammation. (**B**) The pathogenic roles of integrins on VSMCs in AS. The binding of integrins to ligands in ECM can activate the characteristics of VSMCs, including VSMC migration, proliferation, differentiation, and calcification, which eventually leads to vascular stiffness. (**C**) The effects of integrins on immune cells involved in the pathogenic process of AS. The integrins on immune cells can activate several specific signaling pathways, such as PI3K/Akt and Syk/Src, which can induce cascade reactions and further activate the characteristics of immune cells, including migration, proliferation, adhesion, and phagocytosis, thereby causing thrombus formation. (**D**) The specific roles of integrins in platelet in AS pathogenesis. ECM–integrin interactions mainly participate in platelet activation, migration, and secretion, which subsequently causes blood coagulation, thrombosis, and AS plaque instability. (**E**) The effects of integrins on cardiac fibroblasts involved in cardiac fibrosis. The binding of integrins to ECM induces fibroblast activation, migration, and trans-differentiation, which promotes collagen synthesis and deposition. (**F**) The pathogenic roles of integrins on cardiomyocytes in arrhythmias. The activation of cardiomyocytes induced by ECM–integrin interactions mediates myocardial damage and myocardial electrical signal dysfunction, leading to cytoskeletal remodeling. Abbreviations: ECM: extracellular matrix, Rho: Ras homolog family, JNK: C-Jun NH2-terminal kinase, Src: serum creatinine, NF-kB: nuclear factor-k-gene binding, ANXA2: annexin A2, PIP1B: Arabidopsis thaliana aquaporin gene AthH2, FAK: focal adhesion kinase, ICAM-1: intercellular cell adhesion molecule-1, VCAM-1: vascular cell adhesion molecule-1, MMP2: matrix metalloproteinase2, TGF-β1: transforming growth factor-β1, α-SMA: α-smooth muscle actin, ERK1/2: extracellular signal-regulated kinase 1/2, Akt: protein kinase B, OPG: osteoprotegerin, YAP: Yes-associated protein, TAZ: tafazzin, Pyk2: protein tyrosine kinase 2, Ube2n: ubiquitin-conjugating enzyme 2N, Mdm2: mouse double minute 2 homolog, ACE2: angiotensin-converting enzyme 2, Smad 2/3: mothers against decapentaplegic homolog 2/3, Nox4: NADPH oxidase 4, PI3K: phosphoinositide 3-kinase, AMPK: AMP-activated protein kinase, PDIA1-Cit: protein disulfide isomerase A1-citrulline, PDIA1-Arg: protein disulfide isomerase A1-arginine, PAD4: peptidylarginine deiminase 4, CXCL1: chemokine (C-X-C motif) ligand 1, Gα13: guanine nucleotide-binding protein G(13) subunit alpha, mTOR: mammalian target of rapamycin, Vav3: Vav guanine nucleotide exchange factor 3, RhoGAP: Rho GTPase-activating protein, Rac-GEFs: Rac guanine nucleotide exchange factors, c-Src: cellular serum creatinine, PP2A: protein phosphatase 2A, PTEN: phosphatase and tensin homolog, ZO-1: Zona occludens protein 1, Vcl: vinculin, ILK: integrin-linked kinase, Cx43: connexin 43, PyR2: ryanodine receptor 2.

**Figure 3 ijms-25-04096-f003:**
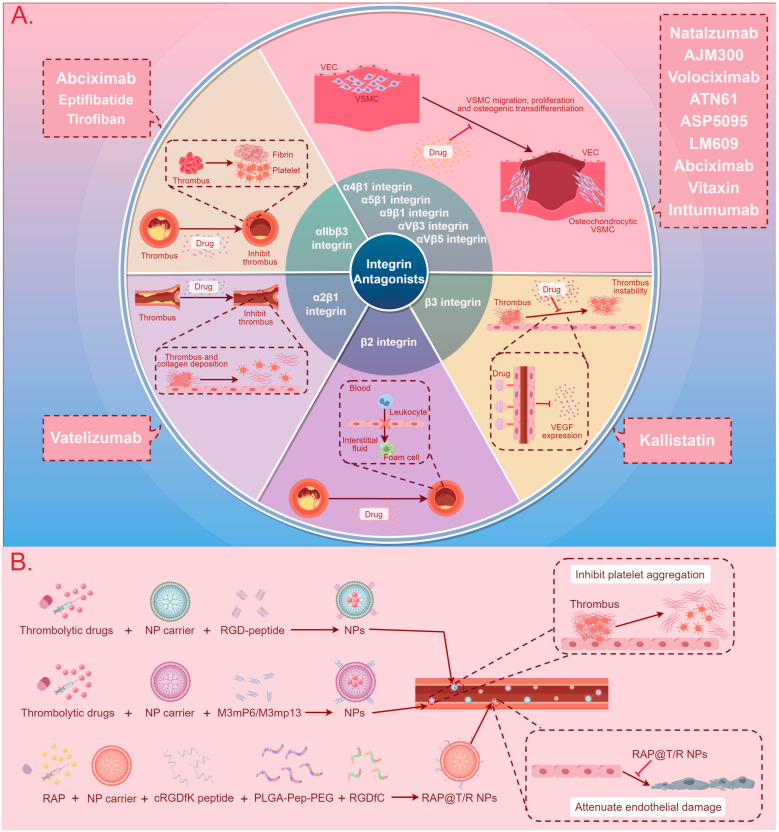
A summary diagram of integrin-based cardiovascular disease treatment methods and therapeutic drugs. (**A**) The use of integrin antagonists to block the binding of integrins with ligands, thereby inhibiting signal pathway transmission and CVD progression. (**B**) The use of NPs to enhance targeting specificity and bioavailability, delay drug release, and mitigate drug toxicity in the treatment of CVDs. Abbreviations: CVD: cardiovascular disease, VEC: vascular endothelial cell, VSMC: vascular smooth muscle cell, VEGF: vascular endothelial growth factor, NP: nanoparticle, RGD-peptide: arginine–glycine–aspartate peptide, RAP: rapamycin, cRGDfK peptide: cyclic arginyl–glycyl–aspartic acid–phenylalanine–lysine peptide, PLGA-Pep-PEG: poly (lactic-co-glycolic acid)–peptide–polyethylene glycol, RGDfC: Arg-Gly-Asp-Phe-Cys, RAP@T/R NPs: RAP targeted and responsive nanoparticles.

**Table 1 ijms-25-04096-t001:** The interaction between integrins on specific cells and ECM components in AS.

Disease	Cell Types	Integrin	ECM	Main Functions	Effects of Signal Modulation	Reference
AS	VEC	α5β1	ANX2 Ox-LDL	Promotes the translocation of α5β1 integrins to lipid rafts and activates the endothelial inflammatory pathway	Foam cell deposition Vascular endothelial inflammation	[35,36]
			COMP	Inhibits the activation of α5β1 integrins	Alleviates the vascular endothelial inflammation	[14]
			Gal-3	Activates the β1 integrin/RhoA/JNK signaling pathway and exacerbates ox-LDL-mediated vascular endothelial injury	Foam cell deposition Vascular endothelial inflammation	[37]
		β4		Activates the Src/NF-kB signaling pathway and promotes the expression of endothelial inflammatory factors	Vascular endothelial inflammation	[38]
		α6β1	CCN1	Activates NF-kB and forms a positive feedback loop with CCN1 and α6β1 integrins	Oxidative stress Vascular endothelial inflammation	[39]
	VSMC	α5β1	MFG-E8	Promotes the MMP2 expression and activates the TGF-β1/Smad2/3 signaling pathway	VSMC calcification Vascular wall calcification	[15]
			α-SMA	Mediates vascular tone dysregulation and VSMC migration	Vasoconstrictive dysfunction	[40]
		β3	Thrombin	Interacts with CD47 and mediates VSMC migration and proliferation	Vascular wall thickening and vascular restenosis	[41]
	Immune cell	αMβ2	ICAM-1	Induces the adhesion and extravasation of immune cells on the vascular endothelium	Vascular endothelial inflammation	[16]
			Gal-9	Activates and promotes the differentiation of monocytes to macrophages	Vascular endothelial inflammation Atherosclerotic plaque formation	[42]
		αDβ2	DHA CEP	Promotes M1 macrophage accumulation in ECM	Vascular endothelial inflammation Atherosclerotic plaque formation	[43]
		αVβ5	Cadherin ICAM-1 E-selectin	Promotes fibronectin expression and macrophage migration	Vascular endothelial inflammation Atherosclerotic plaque formation	[44]
			PPARγ	Promotes M2 macrophage polarization and the expression of anti-inflammatory factors	Attenuates AS and promotes tissue repair	[45]
			Omentin-1	Induces the PI3K/Akt signaling pathway and AMPK phosphorylation	Promotes plaque stability	[46]
			OPN	Attenuates vascular calcification	Promotes positive ischemic neovascularization	[47]
		α4	FABP4	Induces macrophage adhesion	Vascular endothelial inflammation Atherosclerotic plaque formation	[48,49]
		α5β1	EphA2	Promotes immunocyte adhesion		[50]
		β2	E-selectin ICAM-1	Activates the Syk/Src signaling pathway and then promotes the calcium reflux of neutrophils		[51]
				Promotes eosinophil adhesion		[52]
			PAD4	Promotes neutrophil adhesion		[53]
	Platelet	αIIbβ3	Fibrinogen	Activates Rho GTPase RAC1 and RhoA, thereby promoting cytoskeletal reorganization	Thrombus formation	[54]
			GNE495 PF 06260933	Inhibits the pathogenic roles of αIIbβ3 integrins	Inhibits platelet aggregation and clot retraction	[55]
		αVβ3	Fibronectin	Promotes platelet adhesion and aggregation	Thrombus formation	[56,57]

Abbreviations: AS: atherosclerosis, VEC: vascular endothelial cell, VSMC: vascular smooth muscle cell, ANX2: annexin A2, Ox-LDL: oxidized low-density lipoprotein, COMP: cartilage oligomeric matrix protein, Gal-3: galectin-3, JNK: c-Jun N-terminal kinase, Gal-9: galectin-9, CCN1: Cyr61, NF-kB: nuclear factor kappa-B, MFG-E8: milk fat globule-EGF factor 8 protein, α-SMA: α-smooth muscle actin, ICAM-1: intercellular adhesion molecule-1, DHA: docosahexaenoic acid, CEP: 2-(w-carboxyethyl)-pyrrole, E-selectin: endothelial-selectin, PPARγ: peroxisome proliferator-activated receptor γ, OPN: osteopontin, FABP-4: fatty acid-binding protein 4, EphA2: ephrin type-A receptor 2, PAD4: peptidyl arginine deiminase 4, GNE495: Genentech nerve growth factor inhibitor 495, TGF-β1: transforming growth factor β1, ECM: extracellular matrix.

**Table 2 ijms-25-04096-t002:** The interaction between integrins on specific cells and ECM components in cardiac fibrosis.

Disease	Cell Types	Integrin	ECM	Main Functions	Effects of Signal Modulation	Reference
Cardiac fibrosis	Cardiac fibroblast	αVβ5 αVβ3	Latent TGF-β	Activates the TGF-β1/Smad2/3/α-SMA signaling pathway and promotes collagen synthesis	Cardiac fibroblast transdifferentiation and collagen deposition	[76,77]
				Activates the FAK/c-Src/NF-kB signaling pathway and promotes collagen synthesis		[78]
		αVβ1	Latent TGF-β	Activates the FAK/Akt/mTOR and TGF-β/Smad2/3/α-SMA signaling pathways		[79]
			Fibronectin			[80]
			CD63	Promotes the translocation of Smad2/3 and β-catenin, thereby promoting collagen synthesis		[80]
		α2β1	Collagen	Activates FAK and Src	Attenuates collagen deposition	[81]
				Activates PP2A/PTEN signaling pathway and then inhibits Akt and α-SMA expression		[82]

Abbreviations: TGF-β: transforming growth factor β, α-SMA: α-smooth muscle actin, NF-kB: nuclear factor kappa-B, FAK: focal adhesion kinase, Akt: protein kinase B, mTOR: mammalian target of rapamycin, PP2A: protein phosphatase 2A, PTEN: phosphatase and tensin homolog.

**Table 3 ijms-25-04096-t003:** The interaction between integrins on specific cells and ECM components in arrhythmias.

Disease	Cell Types	Integrin	ECM	Main Functions	Effects of Signal Modulation	Reference
Arrhythmias	Cardiomyocyte	β1		Activates the ILK/Akt/Cx43/PI3K/Akt signaling pathway	Diminishes cardiac remodeling and attenuates arrhythmias	[111,112,113]
				Activates the talin/Vcl/ZO-1/Cx43 signaling pathway, thereby promoting Cx43 stability	Stabilizes the myocardial electrical signal Attenuates arrhythmias	[17]
			Fibronectin	Promotes β1 integrin degradation and inhibits RyR2 phosphorylation	Myocardial electrical signal dysfunction	[114]
		α5	Fibronectin	Promotes ECM collagen deposition	Myocardial damage Cytoskeletal remodeling	[115]

Abbreviations: ILK: integrin-linked kinase, Akt: protein kinase B, Cx43: Connexin43, PI3K: phosphoinositide 3-kinase, Vcl: vinculin, ZO-1: zonula occludens-1, RyR2: ryanodine receptor 2, ECM: extracellular matrix.

**Table 4 ijms-25-04096-t004:** The interaction between integrins on specific cells and ECM components in hypertension.

Disease	Cell Types	Integrin	ECM	Main Functions	Effects of Signal Modulation	Reference
Hypertension	PASMC	β3	MMP8	Activates the FAK/YAP/TAZ signaling pathway and promotes PASMC proliferation	Vascular remodeling	[119]
		αVβ3	Hypoxia	Activates the Pyk2/ERK/NF-kB/H_2_O_2_ signaling pathway, thereby reducing PPARγ expression and promoting PASMC proliferation		[120]
				Activates the OPG/FAK/Akt signaling pathway and promotes PASMC proliferation		[61,121]
			OPN	Activates the ERK1/2/Akt signaling pathway and promotes PASMC proliferation		[122]
		β5	PDGF-BB	Activates the Ubal protein/Ube2n/Mdm2/ACE2 and miR-96-5p/mTOR signaling pathways, thereby promoting PASMC proliferation		[123,124]

Abbreviations: PASMC: pulmonary arterial smooth muscle cell, PPARγ: peroxisome proliferator-activated receptor γ, OPN: osteopontin, MMP8: matrix metalloproteinase-8, PDGF-BB: platelet-derived growth factor-BB, FAK: focal adhesion kinase, YAP: Yes-associated prtein, TAZ: PDZ-binding motif, Pyk2: proline-rich tyrosine kinase 2, ERK: extracellular regulated protein kinases, NF-kB: nuclear factor kappa-B, OPG: osteoclastogenesis inhibitory factor, Akt: protein kinase B, Ube2n: ubiquitin conjugating enzyme E2 N, Mdm2: murine double minute2, ACE2: angiotensin-converting enzyme 2, mTOR: mammalian target of rapamycin.

**Table 5 ijms-25-04096-t005:** Integrin-based cardiovascular disease treatment methods and therapeutic drugs.

Category	Related Integrin	Functions	Implication in CVD	Agents in Clinics	Reference
Integrin antagonists and antibodies	αIIbβ3	Inhibits platelet aggregation with fibrinogen and clot retraction	Inhibits thrombosis	Abciximab, Eptifibatide, Tirofiban	[18,133,134,135,136,137]
	α2β1	Inhibits the adhesion of platelets to collagen Inhibits the expression of collagen and collagenase genes Inhibits the phenotypic plasticity of VSMC	Inhibits thrombosis and plaque formation	Vatelizumab	[134,138]
	α4β1	Inhibits the phenotypic plasticity of VSMC	Inhibits plaque formation and platelet aggregation	Natalzumab, AJM300	[139,140,141,142]
	α5β1	Inhibits the phenotypic plasticity of VSMC	Inhibits plaque formation and platelet aggregation	Volociximab, ATN61	[143,144,145]
	α9β1	Inhibits the phenotypic plasticity of VSMC	Inhibits plaque formation and platelet aggregation	ASP5094	[146,147]
	αVβ3	Inhibits the phenotypic plasticity of VSMC	Inhibits plaque formation and platelet aggregation	LM609, Abciximab (c7E3Fab; ReoPro), Vitaxin, Inttumumab,	[143,148,149,150,151,152]
	αVβ5	Inhibits the phenotypic plasticity of VSMC	Inhibits plaque formation and platelet aggregation	LM609 Inttumumab	[151,153]
	β2	Inhibits leukocyte extravasation	Reduces inflammatory tissue damage		[8]
	β3	Inhibits NF-κB nuclear translocation Downregulates VEGF expression	Inhibits angiogenesis	Kallistatin	[154]
Nanotherapy	αIIbβ3	Surfaces with RGD peptide; Loading thrombolytic drugs	Thrombolysis		[19,155]
	αIIbβ3	Contains M3mP6 or M3mp13 Inhibits signaling pathways of αIIbβ3 integrins	Inhibits thrombosis		[23,129,156]
	ανβ3	Contains cRGDfK peptide			[130]
	αVβ3	Releases RAP	Inhibits local inflammation	RAP@T/R NPs	[157]
	α4β1			MMM NPs	[128]

Abbreviations: CVD: cardiovascular disease, VSMC: vascular smooth muscle cell, VEGF: vascular endothelial growth factor, RGD: Arg-Gly-Asp, RAP: rapamycin, MMM: modified macrophage membrane.

## Data Availability

Not applicable.

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
