# Peer review of "Roles of Integrin in Cardiovascular Diseases: From Basic Research to Clinical Implications"

_ijms, 2024, doi:10.3390/ijms25074096_

Round 1

Reviewer 1 Report

Comments and Suggestions for Authors

This review paper deals with an important subject: Intergrin in the cardiovascular system. The review paper is relatively well written; however, some revisions is suggested to improve the paper and make it easy to read for experts and non expert in the field.

1)      The authors use, in many cases, long sentences. This should be avoided, particularly at the beginning of each section.

2)      In several cases, the sentences are referenced. Each sentence and information does not belong to the authors and should have a reference.

3)      The tables are essential but very long, and it is not easy for the reader to understand the information. Thez should divided with specific reasons.

4)      Several important recent papers and reviews, such as therapeutic integrin inhibitors, are not covered ( Slacket al 2022).

5)      The authors did not discuss integrins' crucial interactions with other systems, such as connexin (Okamoto et al., 2021: Amanand Margadant, 2022).

6) The authors should also discuss the importance of glycocalyx and integrin in regulating endothelial and vascular smooth muscle cells (Bkaily and Jacques, 2023; Bkaily et al., 2021.

7)      The authors should carefully check their references and suitability, such as ref 168, which is not in English, and references 51,59,61, and 117 with no authors' names, volume numbers, no year !!!!

Comments on the Quality of English Language

The English could be better

Author Response

  1. The authors use, in many cases, long sentences. This should be avoided, particularly at the beginning of each section.

RE: We apologize that some of the sentences in this review are too long and may make it difficult for readers and peers to read. To make it easier for readers to understand, we changed the long sentences to short sentences and made the logics and meanings of sentences clearer, and also split the sentences at the beginning of each section.

  1. In several cases, the sentences are referenced. Each sentence and information does not belong to the authors and should have a reference.

RE: We noticed that we miss some references in ‘1. Introduction’, ‘2. Potential effects of integrins in CVDs’ and ‘3. Integrin-based therapy’. After combining the reviewers’ recommendations, we addded the corresponding references in the missing places in this review.

  1. The tables are essential but very long, and it is not easy for the reader to understand the information. Thez should divided with specific reasons.

RE: In order to make it easier for the readers to understand the information of the tables, we divided the original Table 1 into 4 tables according to the 4 types of disease discussed in this review, named respectively as ‘Table 1. The interaction between integrins on specific cells and ECM components in AS ’, ‘Table 2. The interaction between integrins on specific cells and ECM components in cardiac fibrosis’, ‘Table 3. The interaction between integrins on specific cells and ECM components in arrhythmias’ and ‘Table 4. The interaction between integrins on specific cells and ECM components in hypertension’, and inserted them into the first paragraph of each type of disease.

  1. Several important recent papers and reviews, such as therapeutic integrin inhibitors, are not covered ( Slacket al 2022).

RE: By combining the expert’s suggestions, we added related content of Slack et al. to this review in lines 846-852 and discussed the application of integrin inhibitors in clinical medicine.

  1. The authors did not discuss integrins' crucial interactions with other systems, such as connexin (Okamoto et al., 2021: Amanand Margadant, 2022).

RE: By combining the expert’s advice, we added the content of integrin interacting with connexin to promote vascular inflammation and atherosclerosis in lines 230-236.

  1. The authors should also discuss the importance of glycocalyx and integrin in regulating endothelial and vascular smooth muscle cells (Bkaily and Jacques, 2023; Bkaily et al., 2021.

RE: According to the reviewer’s advice, we added the discussion of glycocalyx and integrin in regulating endothelial and vascular smooth muscle cells in lines 929-937.

  1. The authors should carefully check their references and suitability, such as ref 168, which is not in English, and references 51,59,61, and 117 with no authors' names, volume numbers, no year !!!!

RE: We apologize for our negligence in the citing literature format. We have deleted the ref 168 in original manuscript and substituted with a new literature in line 1000 and 1004 which is marked red. Also, we checked and modified references 51,59,61, and 117 and the newly modified references are in lines 1146, 1177, 1179 and 1326 respectively.

Reviewer 2 Report

Comments and Suggestions for Authors

Integrins have gained attention in the cardiovascular field in the past decades, mainly in how they can orchestrate cardiac development and homeostasis and modulate injury processes. Zhang et al. compiled the main studies on atherosclerosis, cardiac fibrosis, hypertension, and arrhythmias in this manuscript. Although the manuscript is well written, a few comments need to be addressed.

  • Because the heart is a multicellular organ, and each cell type has its own integrin, the study lacks the general concept of each cell type's expression and how it is dynamically regulated during development and health functions.
  • The CVDs targeted in this manuscript are too broad, and although the immune response is the main final target, each disease is affected by a different cell type. The reviewer understands that it may be difficult to make the text fluid, but the topic of atherosclerosis is very repetitive. The authors did a great job on the cardiac fibrosis section, which must be followed as a good example.
  • Figure 1 must be more integrated within the text.
  • Table 1 was brought too early to the text. Break it per disease or contextualize it better.
  • Figure 2 is missing the integrin isoforms.
  • Although it is a review, some authors' comments/interpretations are missing throughout the text. For example, in the 2.2 section, b1 integrin can act as a pro-collagen (fibrotic) but also as an inhibitor of collagen accumulation. Why does this happen? What are the authors' interpretations regarding these 2 different modes of integrin action?

Minor comments:

  • The pages were wrong-labeled.
  • The order of references was wrong in the text.
  • There is a 2022 updated Circ report for CVDs. 
Comments on the Quality of English Language

No problem was detected. Minor edits

Author Response

  1. Because the heart is a multicellular organ, and each cell type has its own integrin, the study lacks the general concept of each cell type's expression and how it is dynamically regulated during development and health functions.

RE: We adopted the editor's viewpoint and have annotated all the integrins involved in the mechanisms of the 4 diseases with specific cell types, indicating the expression of the corresponding integrins on which cell membranes. In describing the involvement of integrins in the development of diseases, we added the descriptive phrases of dynamic changes in integrins and other molecules, such as "upregulation", "overexpression" and "excessive activation", to demonstrate how integrins dynamically regulate in both health and disease states in general.

  1. The CVDs targeted in this manuscript are too broad, and although the immune response is the main final target, each disease is affected by a different cell type. The reviewer understands that it may be difficult to make the text fluid, but the topic of atherosclerosis is very repetitive. The authors did a great job on the cardiac fibrosis section, which must be followed as a good example.

RE: After combing the reviewer’s suggestion and referring to cardiac fibrosis section,we organized the topic of atherosclerosis according to integrin types, summarizing the involvement of 3 integrin categories in atherosclerosis, including ‘2.1.1 Integrins binding to RGD receptors’, ‘2.1.2 Integrins binding to laminin receptors’, and ‘2.1.3 Integrins binding to leukocyte-specific receptors’. This avoids the repetition of cellular effects when organizing the atherosclerosis section by cell types.

  1. Figure 1 must be more integrated within the text.

RE:We realized that the contents of the Figure 1A and 1B are not mentioned in our manuscript. So we added a description of Figure 1A and 1B to introduction in lines 52-60.

  1. Table 1 was brought too early to the text. Break it per disease or contextualize it better.

RE:According to the editor’s suggestion, we divided Table 1 into 4 tables referring to 4 diseases: atherosclerosis, cardiac fibrosis, hypertension and arrhythmias. The 4 tables are ‘Table 1. The interaction between integrins on specific cells and ECM components in AS’, ‘Table 2. The interaction between integrins on specific cells and ECM components in cardiac fibrosis’, ‘Table 3. The interaction between integrins on specific cells and ECM components in arrhythmias’ and ‘Table 4. The interaction between integrins on specific cells and ECM components in hypertension’.

.

  1. Figure 2 is missing the integrin isoforms.

RE: We have updated Figure 2. We labeled specific integrin isoforms next to the 6 graphics representing integrins.

  1. Although it is a review, some authors' comments/interpretations are missing throughout the text. For example, in the 2.2 section, b1 integrin can act as a pro-collagen (fibrotic) but also as an inhibitor of collagen accumulation. Why does this happen? What are the authors' interpretations regarding these 2 different modes of integrin action?

RE: We apologize for the losing explanation of the different effects of β1 integrins. After combing the reviewer’s suggestion, we added relevant instructions at lines 629 to 631 and 641 to 644 of 2.2 section.

Minor comments:

  1. The pages were wrong-labeled.

RE:This was our negligence. We have modified them to the correct page number.

  1. The order of references was wrong in the text.

RE:We rechecked the references and modified them to the correct order, including the newly added references which are marked in red in our manuscript.

  1. There is a 2022 updated Circ report for CVDs.

RE: We found the 2022 updated Circ report for CVDs and updated our data in the introduction section in lines 34 to 37.

Reviewer 3 Report

Comments and Suggestions for Authors

This present manuscript by Zhang et al summarizes the recent knowledge on the role of integrins in cardiovascular diseases (CVD). The paper covers the most important points related to the integrins and CDVs as the following structure: 1) role of integrins in CVDs (focusing on atherosclerosis, fibrosis, arrhythmias, hypertension), 3) clinical implications of integrins and integrin-based therapies. The authors highlighted that integrins might be both biomarkers and drug targets in CVDs.

The paper is written with smooth language. The main topic of the manuscript is interesting, it could catch the readers’ interest. The strength of it is the large number of references (more than 190) cited and discussed within the manuscript. The authors also included two figures illustrating the structure and the most important integrin-related pathways and two tables on the role of integrins and their clinical implications.

No major concerns were raised regarding the manuscript. Optionally, adding another figure can be considered that summarizes the potential therapeutic targets and drugs (generally the visual form of Table 2).

NOTE: the Reviewer is not a native English speaker, so language check might be recommended.

Author Response

  1. No major concerns were raised regarding the manuscript. Optionally, adding another figure can be considered that summarizes the potential therapeutic targets and drugs (generally the visual form of Table 2).

RE: We created Figure 3 to summarize the names, functions, and implication in cardiovascular disease of integrin-based therapeutic drugs, and visualize the content from Table 2.

  1. NOTE: the Reviewer is not a native English speaker, so language check might be recommended.

RE: Thank you for editor’s suggestion. We have already submitted the manuscript to a language polishing service for optimization.

Reviewer 4 Report

Comments and Suggestions for Authors

The work is devoted to a review of cardiovascular diseases from the point of view of integrin-dependent processes. The review is well written, easy to understand, has a logical narrative, and offers perspectives. There are some problems which are described below. After the comments are taken into account, the article can be published.

1) ‘…paving the way for future innovations in the diagnosis and treatment of CVDs’ (lines 25-26)

- It is worth softening the statement (eg, ‘…which may pave the way…’).

2) ‘In 2019, CVDs remained the foremost cause of mortality worldwide, accounting for an estimated 31% of all global deaths [1-4]’ (lines 33-35)

- References 1, 2 and 3 do not correspond to the statement: 1 - about assessing the effect of a vegan diet for the prevention of CVD; 2 - article published in 2017; 3 - the study was implemented between December 2013 and May 2017.

3) ‘Notably, specific focus has been directed towards dysregulated ligands that target integrins, emphasizing their unique relevance in the context of cardioprotection [6]’ (lines 39-41)

- The article you referred to does not contain any information about therapy.

4) Decipher these abbreviations when first mentioned in the text: ‘RGD’ (line 45), ‘AS’ (line 83), ‘HCAECs’ (line 250).

5) Figure 1 (line 63)

- The color designation in Figure 1A and Figure 1B do not match. In the figure caption, explain the logic of dividing domains into colors.

6) 2. Potential effects of integrins in CVDs (line 94)

- Place the Table 1 and Figure 2 in narrative order.

7) Table 1 (line 100)

- The reference numbers must be in narrative order. Change the reference number order of References so that the numbering of the text references continues in the Table 1.

- Out of the context, the role of the ‘ECM’ column is not entirely clear. Indicate in the title that you are describing the role of interactions of different types of integrins with different ECM components.

8) ‘endothelial microparticles’ (line 241)

- If you use this term, explain its meaning.

9) References (line 746)

- Bring the list of references into a proper form (lines 856, 875-876, 880-881, 1151)

- You listed the same work twice (lines 767-768 and 1129-1130).

Author Response

(The authors gave the same response as above.)
